# Intraspecific Diversity in Aquatic Ecosystems: Comparison between *Spirodela polyrhiza* and *Lemna minor* in Natural Populations of Duckweed

**DOI:** 10.3390/plants11070968

**Published:** 2022-04-01

**Authors:** Manuela Bog, Klaus-Juergen Appenroth, Philipp Schneider, K. Sowjanya Sree

**Affiliations:** 1Institute of Botany and Landscape Ecology, University of Greifswald, D-17489 Greifswald, Germany; manuela.bog@uni-greifswald.de; 2Matthias Schleiden Institute-Plant Physiology, University of Jena, D-07743 Jena, Germany; pilles@gmx.de; 3Department of Environmental Science, Central University of Kerala, Periye 671320, India

**Keywords:** amplified fragment length polymorphism (AFLP), biodiversity, duckweed, intraspecific diversity, *Lemna minor*, Lemnaceae, population analysis, *Spirodela polyrhiza*

## Abstract

Samples of two duckweed species, *Spirodela polyrhiza* and *Lemna minor*, were collected around small ponds and investigated concerning the question of whether natural populations of duckweeds constitute a single clone, or whether clonal diversity exists. Amplified fragment length polymorphism was used as a molecular method to distinguish clones of the same species. Possible intraspecific diversity was evaluated by average-linkage clustering. The main criterion to distinguish one clone from another was the 95% significance level of the Jaccard dissimilarity index for replicated samples. Within natural populations of *L. minor*, significant intraspecific genetic differences were detected. In each of the three small ponds harbouring populations of *L*. *minor*, based on twelve samples, between four and nine distinct clones were detected. Natural populations of *L. minor* consist of a mixture of several clones representing intraspecific biodiversity in an aquatic ecosystem. Moreover, identical distinct clones were discovered in more than one pond, located at a distance of 1 km and 2.4 km from each other. Evidently, fronds of *L. minor* were transported between these different ponds. The genetic differences for *S. polyrhiza*, however, were below the error-threshold of the method within a pond to detect distinct clones, but were pronounced between samples of two different ponds.

## 1. Introduction

Duckweeds belong to the monocotyledonous aquatic plant family Lemnaceae [1], comprising 36 species [2,3,4]. Although duckweeds can bear flowers, fruits and seeds, the most common mode of their reproduction is vegetative propagation. Daughter fronds arise from a mother frond by budding out of pockets [5]. The daughter fronds that have a common ancestral mother frond, together with their progeny, form a clone. This vegetative propagation proceeds with the fastest growth rates known in Angiosperms [6,7]. The two species, investigated in the present study, *Spirodela polyrhiza* (L.) Schleid. and *Lemna minor* L., have clone-specific doubling times of approximately 2.3 and 1.7 days, respectively. In laboratory experiments, both *S. polyrhiza* and *L*. *minor* were reported to flower [8,9]. In the present study, however, we have never observed the two species flowering, neither in the laboratory nor outdoors. Thus, clonal propagation is by far the dominating mechanism of propagation. However, in *S. polyrhiza*, sexual reproduction is assumed to be not too infrequent at the population level because of the very high number of individuals in natural populations. This was concluded by Ho et al. [10] as effects of recombination in their study. During the past few years, it became clear that some of the physiological properties like growth rate [6,7], turion formation capacity [11], protein content [12], and starch accumulation capacities [13,14,15] vary between clones that belong to the same species. This raises the question whether a natural population of duckweed constitutes only a single clone, or whether several clones coexist. It is a pertinent ecological issue bearing consequences on both the basic science and the applications of duckweeds [16,17], wherein proper collection, isolation and maintenance of a clone becomes a prerequisite for reproducible results. Hence, this question led us to investigate the probable existence of clonal diversity or intraspecific diversity of the predominantly vegetative propagating species *S. polyrhiza* and *L. minor* in a pond ecosystem.

Because of the reduction in morphological and anatomical structures, differentiating the species of Lemnaceae on a morphological basis [3] is by itself a difficult task, even for highly specialized experts. Over the last fifteen years, notable progress has been made in the field of molecular taxonomy of duckweeds by employing different techniques including the use of nuclear and plastid markers [18,19,20,21,22,23,24,25] (for a review, see Bog et al. [26]). However, the characterization of intraspecific genetic variations, i.e., distinct identification of clones within a given species is still at its infant stage [27]. Most recent efforts in this direction are genotyping clones of *S. polyrhiza* by sequencing NB-ARC-related genes [28] and application of genotyping-by-sequencing [29], or even SSR markers [30,31]. Nevertheless, cross-species amplification for NB-ARC-related genes and Simple sequence repeat (SSR) markers is low within duckweeds [28,31].

The traditional definition of a duckweed clone refers to fronds that have originated from a common mother frond (ancestor) by vegetative propagation. The samples used in the present study were collected from natural populations, in which case, it was not possible to evaluate on a morphological basis whether the two samples had originated from a common mother frond or not; in other words, whether the samples belong to the same clone. Therefore, in order to carry out the present study on natural populations of the same species, it was necessary to employ molecular methods and, consequently, a molecular definition of the term clone had to be derived. In the present study, we suggest the molecular criterion that, in order to be considered as a clone, a sample should be distinguishable from another sample by molecular methods, which should be confirmed by statistical validation to be significantly above the experimental error.

In an initial effort to characterize intraspecific differences, Bog et al. [18,19,20,21,22,23,24,25] used the method of amplified fragment length polymorphism (AFLP) and detected, in many cases, clear intraspecific differences. Moreover, it was shown that this method is clearly superior to plastidic barcoding [26]. Therefore, we chose AFLP as an inexpensive method to analyse the intraspecific or clonal diversity of duckweed populations. This makes it possible to address the question of whether populations of a single species existing in small ponds of only a few hundred square meters, where duckweeds can be found frequently, are homogenous concerning the intraspecific diversity, i.e., comprising only a single clone, or if several clones of a species coexist in the same pond. Cole and Voshkuil [32] reported, for the first time, the co-existence of several clones of duckweed (*L. minor*) in natural populations. These authors investigated 285 fronds in total, from 11 pond populations (i.e., on average 26 fronds per pond) by allozyme variations resolving 16 putative loci. In the present project, we used AFLP because this method has much higher resolving power than allozyme variation (see Table 1). To the best of our knowledge, this is the first report of characterization of intraspecific variations within a natural population of duckweeds directly at the level of DNA.

## 2. Results

### 2.1. Evaluation of AFLP Fingerprinting

AFLP fingerprinting of 2 × 12 clones of *S. polyrhiza* yielded a total number of 81 reproducible fragments for the four primer combinations: (i) 26 loci (118–564 bp), (ii) 21 loci (100–498 bp), (iii) 20 loci (102–420 bp), and (iv) 14 loci (117–460 bp); that of 3 × 12 clones of *L. minor* yielded a total number of 133 reproducible fragments: (i) 48 loci (111–546 bp), (ii) 40 loci (100–498 bp), (iii) 18 loci (113–384 bp), and (iv) 27 loci (115–411 bp). The Euclidean error rate, calculated from five parallel doublet runs of samples for each species, resulted in 1.5% for the data set of *S. polyrhiza*, and 1.7% for *L. minor*. Table 1 shows characteristic band statistics and diversity indices for the AFLP data sets. The high proportion of polymorphic bands suggests a higher genetic variability of samples from the ponds of *L. minor* than ponds of *S. polyrhiza*. This assumption is strengthened by the Shannon index, which is very low for the respective *S. polyrhiza* ponds (0.9–2.2) in comparison to *L. minor* ponds (16.2–28.9). Additionally, AMOVA shows that there is 99.2% of genetic variability between the two *S. polyrhiza* populations while it is 0.8% within the populations, which leads to a *Phi_ST_* value of 0.992. For *L. minor*, there is 24.1% of genetic variation among the three populations and 75.9% within the populations, leading to a *Phi_ST_* value of 0.241.

### 2.2. Average-Linkage Cluster Analysis

Results of AFLP analysis were presented by dendrograms. For the statistical evaluation of the results, a methodological error-threshold was calculated based on the investigated replicates. The grey bars in the dendrograms (Figure 1) indicate the mean dissimilarity and its 95% confidence interval of the replicates. Therefore, only the branches to the left side of the vertical grey bar represent separate clones. The other branches were caused, e.g., by erroneous AFLP bands on the gel and were below the threshold level. For *S. polyrhiza* (Figure 1a), the upper confidence interval at approximately 0.08 Jaccard dissimilarity units cuts the dendrogram in such a way that only the separation into samples from two different places of origin were accepted as distinct clones (Spirodela1, Spirodela2; see Figure 1a and Table 2). No genetic differentiation was detected between samples of the same origin. These data are based on the analysis of 2 × 12 collected samples out of several millions of fronds in the ponds.

The total 36 samples of *L. minor* (out of several hundred million fronds in the ponds during the peak time) were separated into a larger number of groups out of the confidence interval (Figure 1b), and finally into 20 groups. Two main branches separated all 36 investigated samples into two large groups of equal numbers. At the bottom of the dendrogram there were three samples, i.e., loL01, 02 and 05. These three samples have to be considered as one single, distinct clone, termed as Lemna01 (see Figure 1b and Table 2). It follows sample loL06 in the tree (Figure 1b), forming a distinct clone, Lemna02. The next group is represented by the samples loL10, 11, and 12, forming the distinct clone, Lemna03. All clones mentioned until now are from the pond in Lotschen. The remaining part of the lower branch of the dendrogram consisted of 14 samples, all of them were separated into three distinct clones, i.e., clone Lemna04: schl01, 03, gro09, 12; clone Lemna05: sch06, 07; clone Lemna06: gro01–04, 07, 08, 10, 11. It should be stressed that clone Lemna04 has representatives in the pond at Groeben, as well as in the pond at Schloeben. The upper branch of the dendrogram starts with ten samples, subdivided by this analysis (Figure 1b) into nine different, distinct clones (Lemna07–Lemna15). Only samples sch09 and sch11 formed a common distinct clone, i.e., Lemna07. All other samples formed single distinct clones themselves, originating mainly from Schloeben (Lemna08, Lemna09, Lemna10, Lemna11, Lemna14, Lemna15), and two of them also from Groeben (Lemna12, Lemna13) (Figure 1b). The upper part of the dendrogram contained five samples that had originated from Lotschen, forming the distinct clones Lemna16, Lemna17, Lemna18, Lemna19 and Lemna20. The 20 clones of *Lemna* were consequently classified as distinct clones (Table 2).

## 3. Discussion

### 3.1. No Clonal Diversity Could Be Detected within Natural Populations of Spirodela polyrhiza 

The populations of *S. polyrhiza* in the two ponds in Moscow and Lotschen could be easily distinguished from one another, indicating the existence of two distinct clones of this species. This is in agreement with the results of Xu et al. [33], who reported whole genome sequencing of 68 world-wide collected clones of *S. polyrhiza*, including clone 9509 from Lotschen (see also [34]) and 9511 from Moscow, collected from the same ponds as the AFLP samples analysed here. These authors reported high similarity of both clones (belonging to the “European population”), but both clones from Lotschen and Moscow were distinguished. The first detailed comparison between the genomes of two *S. polyrhiza* clones (7498 from USA and 9509 from Lotschen, Germany [34,35]) demonstrated also very small differences. Similar conclusions about low genetic variation within *S. polyrhiza* were drawn by Ho et al. [10] on the basis of 38 clones of *S. polyrhiza*, almost all of them from Northern America, as well as from Bog et al. [20] and Feng et al. [25], where more than 40 defined *S. polyrhiza* clones, selected from world-wide stock collections, were used in each study. In the present context, it is important that the two investigated populations of *S. polyrhiza* were probably formed by only one genotype each. This was concluded on the basis of the low fraction of investigated plants with 2 × 12 samples from the two ponds using AFLP analysis. We evaluated the number of fronds of the total population in the pond in Lotschen to ca. 10^7^. Such a large population can hardly be investigated in detail. The genome-wide mutation rate in *S. polyrhiza* was found to be within the range of mutation rates reported for unicellular eukaryotes and Eubacteria, but was more than seven times lower than the reported rates for multicellular eukaryotes [33]. Further, Xu et al. [33] found a much slower decay of the linkage disequilibrium with physical distance between linked loci for the “European population” compared to the “Southeast Asian population”, and concluded that sexual reproduction might therefore be less frequent in the first mentioned population contrary to the latter one. It therefore could be concluded that low mutation rates and predominantly asexual reproduction might have led to the lack of genotype diversity in the two investigated populations from ponds in Lotschen and Moscow, both belonging to the “European population”. As this holds true for samples from all over the world [33], environment might not have an essential influence on the mutation rate—in contrast to the flower frequency.

### 3.2. Natural Population of Lemna minor Comprises of Several Distinct Clones

The situation was quite different in three populations of *L. minor*, with conclusions based on the same number of investigated random samples per pond. It is impressive to note that in all the investigated natural populations existing in small ponds of few hundred square meter surface area, several distinct clones of *L. minor* co-existed. As in the case of *S. polyrhiza*, populations of *L. minor* could possess very high surface coverage rates. Despite investigating only a small fraction of this huge population, AFLP results demonstrate that the genetic variations within the species *L. minor* must be higher than in *S. polyrhiza*. It should be mentioned here that the genome of *L. minor* is approximately three to four times larger than that of *S. polyrhiza*, and that it shows much higher genomic variation as it is reflected, e.g., in its genome sizes between clones of this species compared to those of *S. polyrhiza* [20,36], which could be a possible explanation of the existence of a larger number of distinct clones. On the other hand, the higher mutation rate of *L. minor* in comparison to *S. polyrhiza* may be proposed; the reason for this is not known at the moment. Unfortunately, it is not yet possible to compare the genomic differences between clones of *L. minor* directly as presently only one high-quality draft genome is available (Rob Martienssen, www.lemna.org (accessed on 12 February 2022); for a review see [37]). A higher rate of sexual reproduction could be another possible explanation, but no quantitative data are available to compare the two species in this point under natural conditions. However, very recently, the existence of hybrids between *L. minor* and *L. turionifera* [38], and between *L. minor* and *L. gibba* [39], was demonstrated by molecular analysis (tubulin-based polymorphism), suggesting sexual propagation. Such hybrids are not known in *S. polyrhiza*, which might be a hint that sexual propagation in this species is rarer.

Tang et al. [40] investigated *S. polyrhiza*, *Landoltia punctata* (G. Mey.) Les & D.J.Crawford, and three *Lemna* species from Lake Tai (2250 km^2^ large) using cpDNA markers. The authors could easily identify the duckweed species using this method, but could not detect any intraspecific differences. El-Kholy et al. [41] compared the genetic diversity between populations and collected samples from the Nile delta, Egypt, using the fingerprinting method of inter-simple sequence repeats (ISSRs). They detected intraspecific differences within *L. minor* and *L. gibba* L.; however, in both cases, the large size of the water bodies questions the existence of only a single population of duckweed in each of these study areas. Paolacci et al. [42] collected clones of *L. minor* from a restricted geographic area, i.e., the southern part of Ireland, and detected clearly intraspecific genetic variability by AFLP—as for the invasive species *Lemna minuta*. However, in this case also we cannot consider the samples as belonging to a single population. Hence, these results are rather similar to those of clones from stock collections from all over the world, implying geographical influence on different populations (for recent reviews, see [26]).

### 3.3. Natural Transport of Clones between Ponds

One distinct clone of *L. minor*, Lemna04, was detected in both lakes situated in Groeben and Schloeben, substantiating that there should have been a transfer of fronds either from one pond to the other, or an identical clone from a distant pond to the two ponds in Groeben and Schloeben, which is less probable. The distance between the two ponds is only 1 km. Thus, epizoochorous transport by water birds, but also by amphibians or rodents, seems to be possible. Beside mutation and sexual reproduction, both very rare processes in duckweeds, epizoochorous transport from other ponds seems to be a good candidate to explain the intraspecific variability in *L. minor*. The direction of the putative transfer is unknown as we had detected two samples of this clone in both the ponds. It has already been reported by Coughlan et al. [43,44,45] that duckweeds can be transported over certain distances by Mallard ducks, especially frequently over small distances [44]. The present results demonstrate the possible outcome of such natural transport between ponds, i.e., an increase in the intraspecific diversity of populations. However, this does not explain the different genetic variability between *S. polyrhiza* and *L. minor*. M. A. K. Jansen, University College Cork, Ireland, wrote: ”The issue is that dispersal (of duckweeds) relies on entanglement between feathers, sticking to the bird, survival of drought stress during flight, and release upon arrival in a new water body. On balance, considering all these parameters, there are no clear differences between species” (personal communication to K.J.A.).

## 4. Materials and Methods

### 4.1. Taxon Sampling and Cultivation of Clones

Duckweed samples were collected from five populations in four ponds (Table 3). The circumference of each pond was divided in 12 equidistant sampling sites, and 12 samples (colonies) were collected from each population at these sites. Three of the ponds were in villages in Thuringia, Germany (Lotschen, Groeben, Schloeben; Figure 2) and one distant pond in Moscow, Russia (Table 3). The distance from Lotschen (“Ruttgersdorf-Lotschen”) to Groeben and Schloeben is approximately 2.4 km and 1.9 km, respectively, from Groeben to Schloeben is approximately 1.0 km, and from Lotschen to Moscow is approximately 2000 km.

We extrapolated the density of a *S. polyrhiza* population by counting a square of 10 × 10 cm^2^ in the pond of Lotschen to be ca. 6 × 10^4^ m^−2^, resulting in a total population of ca. 10^7^ in the pond during the peak of the summer season. In the case of *L. minor*, Hicks [46] reported 10^6^ fronds m^−2^.

The late Elias Landolt, ETH Zurich, Switzerland, confirmed the species identities of the 3 × 12 *L. minor* samples and 2 × 12 *S. polyrhiza* samples using morphological markers. Thus, from each pond and each population, 12 samples were collected, and single colonies were sterilized [47]. All samples were tested in nutrient medium supplemented with 25 mM glucose for the absence of microbial contamination. Single colonies have different numbers of fronds connected by stolons, and all these fronds originate from a single mother frond. Offspring of a colony are therefore clonally-related fronds, and the progeny of a single colony each were used for analysis—called a “sample” in the present paper. The samples were cultivated under axenic conditions in continuous white light (100 µmol m^−2^ s^−1^) at 25 ± 1 °C in N-medium [48]: 8 mM KNO_3_, 0.15 mM KH_2_PO_4_ (increased in comparison to the original protocol), 1 mM MgSO_4_, 1 mM Ca(NO_3_)_2_, 5 µM H_3_BO_3_, 0.4 µM Na_2_MoO_4_, 13 µM MnCl_2_, and 25 µM Fe(III)NaEDTA, in order to get sufficient plant material from the collected clonal samples. In most cases, plants were harvested after 14 days of cultivation, frozen in liquid nitrogen, and stored at −80 °C for further use. One sample from each of the two *S. polyrhiza* populations were used for whole genome sequencing, as clone 9509 from Lotschen and 9511 from Moscow [33,34].

### 4.2. DNA Isolation and AFLP Analysis

Total DNA was isolated immediately after grinding in liquid nitrogen using the Cetyl Trimethyl Ammonium Bromide (CTAB) method, following the protocol of Doyle and Doyle [49]. DNA was quantified by a NanoVue spectrophotometer (GE Healthcare Europe GmbH, Freiburg, Germany) at 260 nm. The complete AFLP procedure, mainly according to the protocol of Vos et al. [50], as described by Bog et al. [19], was followed. Infra-red dye (IRD)-labelled primers were used for the selective PCR amplification that consequently labelled the electrophoretic bands. After testing a large number of primers, the following four primer combinations were selected for AFLP analysis: (i) *Eco*RI-ATT/*Mse*I-CAC, (ii) *Eco*RI-ATT/*Mse*I-CAT, (iii) *Eco*RI-ATT/*Mse*I-CCA, (iv) *Eco*RI-ATT/*Mse*I-CTA [15,16]. An automated DNA sequencer (model 4000 L; Li-Cor Biosciences, Bad Homburg, Germany) was used for electrophoretic separation and detection of generated fragments [51]. AFLP patterns were manually compiled into a 0/1-matrix (“1” for presence, “0” for absence of a band), assuming that bands of equal fragment size are homologous and represent independent loci. This matrix for the investigated clones has been made available as a Appendix A of the present paper (Appendix A). As a measure for reproducibility, the Euclidean error rate was assessed by making two parallel, independent preparations for five of the samples for each of the two investigated species. These replicates were not included in the final evaluation.

### 4.3. Data Analyses

We used “Cluster Analysis” as the method, and the confidence interval as a statistical measure. To avoid confusion, we designated the clones that were collected, propagated and maintained from a single colony of the natural population, as “sample” (we investigated 5 × 12 = 60 such clonal samples) and based on the molecular method, the genetically diverse samples of the same species were categorized and characterized as “distinct clones”.

For AFLP, we calculated measures for band statistics, like mean band presence per sample, number of polymorphic bands, number of fixed bands (=band presence in all samples of a pond), number of private bands (=band presence in at least some clones of a pond), and number of fixed private bands (=exclusive band presence in all samples of a pond). Additionally, we calculated Jaccard dissimilarity matrices for the two data sets, *S. polyrhiza* and *L. minor*, that were used in subsequent analysis. An analysis of molecular variance (AMOVA), *Phi_ST_* (analogue value of *F_ST_*) and Shannon’s index were calculated to evaluate the molecular variation and diversity within our data sets. All aforementioned calculations and analyses were done using the program FAMD v1.31 [52], which can also handle missing data. To check the clones that could be characterized as independent AFLP phenotypes (“distinct clones”), we performed an average-linkage cluster analysis based on the Jaccard dissimilarity matrices in R v. 3.2.3 [53]. Additionally, the mean Jaccard dissimilarity based on the replicates and its 95% confidence intervals were calculated, which were used as threshold values for characterization of distinct clones.

## 5. Conclusions

A remarkable intraspecific biodiversity exists in natural populations of *Lemna minor*, but not in populations of *Spirodela polyrhiza*, suggesting either a higher mutation rate or a higher rate of sexual reproduction in *Lemna minor*.The intraspecific biodiversity in the ponds was further enhanced by the putative transfer (most probably epizoochorous by birds) of plants between closely spaced ponds.The high intraspecific diversity of *Lemna minor* may have a role in the adaptation of the natural duckweed populations to the changing environmental conditions.

## Figures and Tables

**Figure 1 plants-11-00968-f001:**
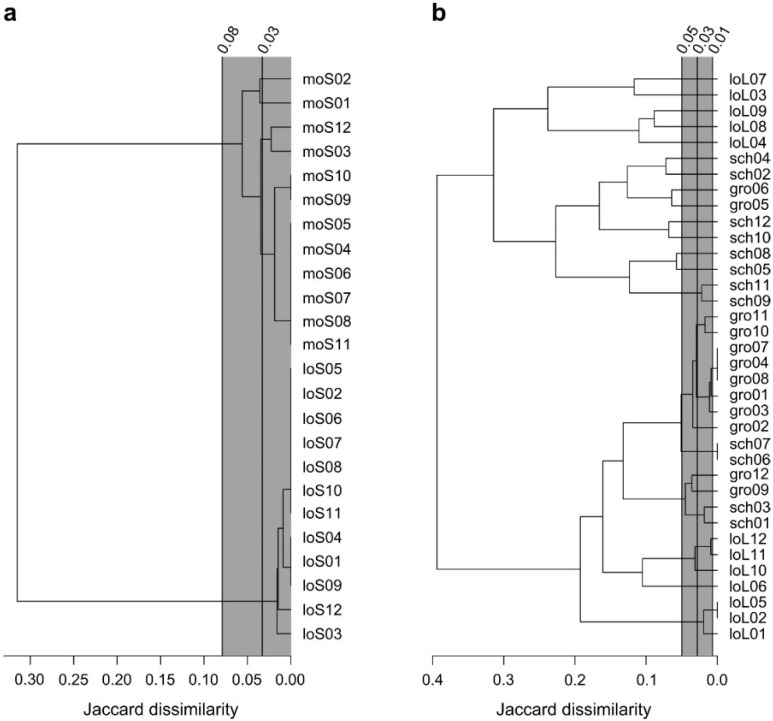
Average-linkage cluster analysis based on Jaccard dissimilarities for (**a**) *Spirodela polyrhiza* and (**b**) *Lemna minor*. Grey bar indicates mean Jaccard dissimilarity and its 95% confidence interval of the replicates. All samples separating to the left of the grey bar can be considered as different clones. The absolute Jaccard values are given at the top. lo = Lotschen, mo = Moscow, gro = Groeben, sch = Schloeben. S = *Spirodela polyrhiza*, L = *Lemna minor*. All samples from gro and sch are *Lemna minor*.

**Figure 2 plants-11-00968-f002:**
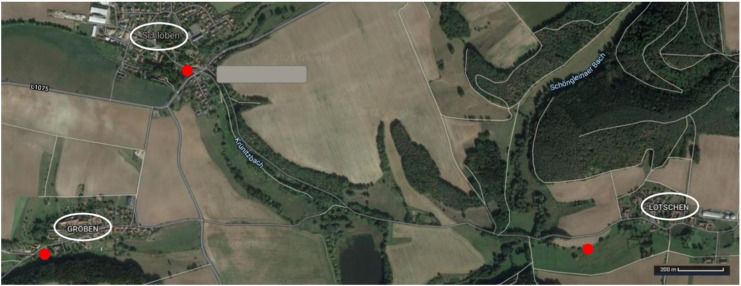
Map of the three closely located ponds in villages in Thuringia, Germany (Lotschen, Groeben, Schloeben), represented by red dots.

**Table 1 plants-11-00968-t001:** Band statistics of AFLP analysis for the investigated ponds. lo = Lotschen, mo = Moscow, gro = Groeben, sch = Schloeben, S = *Spirodela polyrhiza*, L = *Lemna minor*, SD = standard deviation.

	*S. polyrhiza*		*L. minor*		
	moS	loS	loL	Gro	sch
mean band presence per sample (mean ± SD)	53 ± 5	72 ± 7	92 ± 17	107 ± 15	94 ± 14
number polymorphic bands	7	3	70	56	61
number fixed bands	52	73	53	68	67
number private bands	5	22	4	0	4
number fixed private bands	2	19	0	0	0
Shannon’s index	2.2	0.9	28.8	16.2	28.9
*Phi_ST_*	0.992		0.241		

**Table 2 plants-11-00968-t002:** Distinct clones of the species *Spirodela polyrhiza* and *Lemna minor* and their origin by collected samples from ponds in Moscow, Lotschen, Groeben and Schloeben. For further explanations, cf. Table 1.

Distinct Clones	Moscow (mo)	Lotschen (lo)	Groeben (gro)	Schloeben (sch)
Spirodela1	moS01–12			
Spirodela2		loS01–12		
Lemna01		loL01, 02, 05		
Lemna02		loL06		
Lemna03		loL10–12		
Lemna04			gro09, 12	sch01, 03
Lemna05				sch06–07
Lemna06			gro01–04, 07–08, 10–11	
Lemna07				sch09, 11
Lemna08				sch05
Lemna09				sch08
Lemna10				sch10
Lemna11				sch12
Lemna12			gro05	
Lemna13			gro06	
Lemna14				sch02
Lemna15				sch04
Lemna16		loL04		
Lemna17		loL08		
Lemna18		loL09		
Lemna19		loL03		
Lemna20		loL07		

**Table 3 plants-11-00968-t003:** Details of the location of ponds selected for investigation, duckweed species and samples collected. For further explanations, see Table 1.

Species	Location	Samples	Pond Size
*Lemna minor*	Groeben, Thuringia, Germany 50°53′06″ N 11°40′53″ E	gro01–gro12	900 m^2^
*Lemna minor*	Schloeben, Thuringia, Germany 50°53′32″ N 11°41′26″ E	sch01–sch12	400 m^2^
*Lemna minor*	Lotschen, Thuringia, Germany 50°53′07″ N 11°42′56″ E	loL01–loL12	700 m^2^
*Spirodela* *polyrhiza*	Lotschen, 50°53′07″ N 11°42′56″ E	loS01–loS12	700 m^2^
*Spirodela* *polyrhiza*	Botanical Garden Moscow, Russia 55°50′36″ N 37°35′23″ E	moS01–moS12	100 m^2^

## Data Availability

All data are available in the manuscript and Appendix A.

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
