# Peer review of "Intraspecific Diversity in Aquatic Ecosystems: Comparison between Spirodela polyrhiza and Lemna minor in Natural Populations of Duckweed"

_plants, 2022, doi:10.3390/plants11070968_

Round 1

Reviewer 1 Report

So there was 2 questins to be answered in the ms:

  1. Whither the populations of S. polyrhiza and L. minor consists of several independent clones
  2. wheither there is a possibility to distinguish the clones based on AFLP markers.

First question is rather simple and the expectation is that there could be a number of clones in apopulation because of repeated passive dispersal, or sexual reporduction. And previous studies have shown that population of L. minor could indeed be formed by a number of independent clones. While no such information was available for Spirodela. AFLP results also showed the same. To that end, this result is hardly sound. There is an interesting question that might be discussed in addition. What the difference between Spirodela and Lemna can be used to explain this different pattern? I believe the ducks or others can transport individuals of both species from pond to pond. 

A second question also yelded the expected results. Well, if allozyms differentiated clones within populations, then AFLP markers would do it as well. Here, I think it can be interesting to give any indication wheither the AFLP performed better than the allozymes or is there any reason for other methods worse to develop to distinguish the clones? 

L77:81 - actually belongs to methods section and does not fit well the introduction part. I suggest ot considere the removing it from here.

L212: S. polyrhiza must be in italics

Author Response

So there was 2 questins to be answered in the ms:

  1. Whither the populations of S. polyrhiza and L. minor consists of several independent clones
  2. wheither there is a possibility to distinguish the clones based on AFLP markers.

First question is rather simple and the expectation is that there could be a number of clones in apopulation because of repeated passive dispersal, or sexual reporduction. And previous studies have shown that population of L. minor could indeed be formed by a number of independent clones. While no such information was available for Spirodela. AFLP results also showed the same. To that end, this result is hardly sound.

Reply: Although it seems a rather simple question and according to the expectation, there is only a single publication (Cole and Vunsh, 1996) demonstrating the existence of different clones of duckweeds within the same population and this was done on the basis of allozyme investigations.

With respect to the possibility to distinguish clones based on AFLP markers, we have clearly demonstrated it in case of L. minor that several distinct clones exist in each of the investigated natural population.

There is an interesting question that might be discussed in addition. What the difference between Spirodela and Lemna can be used to explain this different pattern? I believe the ducks or others can transport individuals of both species from pond to pond.

Reply: We extended the discussion of possible different reasons. We mentioned before that lower genomic stability in L. minor might be one reason. We added now that different flowering frequency might be another reason (cf. chapter 3.2). We extended also the discussion of the possibility that L. minor could be easier transported, e.g. by ducks but we came to the conclusion that this mechanism does not explain the differences between the two species (chapter 3.3).

A second question also yelded the expected results. Well, if allozyms differentiated clones within populations, then AFLP markers would do it as well. Here, I think it can be interesting to give any indication wheither the AFLP performed better than the allozymes or is there any reason for other methods worse to develop to distinguish the clones? 

Reply: We selected the method of AFLP for our investigations because this method shows much more resolution power.  We cited the number of putative loci for the allozyme method and mentioned now that this has to be compared with the results given in Table 1 (Introduction of the revised version). This shows clearly the higher resolution of AFLP compared with allozyme investigations.

L77:81 - actually belongs to methods section and does not fit well the introduction part. I suggest ot considere the removing it from here.

Reply:  We did it. Thanks.

L212: S. polyrhiza must be in italics

Reply:  We corrected this mistake.

Reviewer 2 Report

The manuscript entitled “Intraspecific diversity in aquatic ecosystems: Comparison between Spirodela polyrhiza and Lemna minor in natural populations of duckweed” by Bog et al. provides some details of infra and interpopulation variability of two sympatric duckweed species. The general concept is very interesting and worthy of future exploration, but in my opinion presented results have preliminary character.

Major issues:

  1. Limited sampling - The sampling as for population studies is quite limited and I’m not convinced that sampling strategy is adequate for such study, as for each station samples from one square  dm were sampled. 
  2. Marker systems - I really can’t find any justification of marker selection. The specific SSR were for both genera were developed over three years ago, which are far superior than ALFP used in this study: https://bsapubs.onlinelibrary.wiley.com/doi/10.1002/aps3.1153

Codominant markers enable performing more population genetic analyses and specific SSR rolls out contamination problems, which is especially common in water plants, which are host for many eucaryotic and procaryotic microorganisms. 

Also the authors didn’t explain how they managed with possible contamination, which could be amplified by non-specific AFLP primers.

  1. Data analysis - Since the chosen marker system limits potential of some analyses, even some basic for population genetic methods are missed like STRUCTURE or clustering bootstrap values not based on genetic similarities. Some more general analysis can be done, like NJ clustering of the whole dataset, which could also show separateness of both species. Study of adaptation is also some way to go, but the loci neutrality wasn’t analyzed too. 

Author Response

The manuscript entitled “Intraspecific diversity in aquatic ecosystems: Comparison between Spirodela polyrhiza and Lemna minor in natural populations of duckweed” by Bog et al. provides some details of infra and interpopulation variability of two sympatric duckweed species. The general concept is very interesting and worthy of future exploration, but in my opinion presented results have preliminary character.

Major issues:

    Limited sampling - The sampling as for population studies is quite limited and I’m not convinced that sampling strategy is adequate for such study, as for each station samples from one square  dm were sampled.

Reply: If the reviewer mentioned the number of samples per pond – how many samples should be taken to cover the millions of fronds even in a small pond of a few hundred square meters? We think that our conclusions were very carefully drawn and were covered by the number of investigated samples. If the reviewer means the sampling strategy, we tried to make clear in the revised version that we did not collected plants from 1 dm2. Instead we collected one single colony (containing clonal fronds of the same mother frond) per sampling point. Collecting the plants from 1 dm2 would most probably result in a mixture of several clones.

    Marker systems - I really can’t find any justification of marker selection. The specific SSR were for both genera were developed over three years ago, which are far superior than ALFP used in this study: https://bsapubs.onlinelibrary.wiley.com/doi/10.1002/aps3.1153

Reply: SSR markers are critical with cross-amplification and usually need much more establishment time for certain species. The successful amplification within a family is very low as stated in Fu et al. 2020. The cross-amplification for the developed markers by Xu et al. 2018 was also low and yielded only monomorphic loci, which would not have been helpful. Since cross-amplification within genera might be a problem too, we decided for AFLP not to have different genotyping methods for our study organisms and to have better comparability between them.

Codominant markers enable performing more population genetic analyses and specific SSR rolls out contamination problems, which is especially common in water plants, which are host for many eucaryotic and procaryotic microorganisms. Also the authors didn’t explain how they managed with possible contamination, which could be amplified by non-specific AFLP primers.

Reply:  In the revised manuscript we made clearer that all plants were sterilized and the success of sterilization was tested by cultivating the plants after sterilization in glucose-containing nutrient medium. Also all media and glass ware was used only after sterilization and the transfer was always carried out in a safety box. Therefore, we can exclude a contribution of eucaryotic or procaryotic contaminations.

    Data analysis - Since the chosen marker system limits potential of some analyses, even some basic for population genetic methods are missed like STRUCTURE or clustering bootstrap values not based on genetic similarities. Some more general analysis can be done, like NJ clustering of the whole dataset, which could also show separateness of both species. Study of adaptation is also some way to go, but the loci neutrality wasn’t analyzed too.

Reply: Additionally, to the presented average-linkage clustering based on Jaccard dissimilarity, we also tested the methods of structure analysis and Principal Coordinate Analysis (PCA). Both methods had much lower resolution power and resulted in a lower number of defined taxonomic groups. Moreover, it was difficult to characterize these groups by statistical methods as it was possible for average-linkage clustering. Therefore, we did not include these methods in the manuscript. Separating the two species from each other is easily possible with AFLP but the present manuscript is focused on intraspecific differences.

The sampling design is not suitable for a study of adaptation. Usually, programmes searching for putative loci under selection require populations to be in Hardy-Weinberg-Equilibrium. Since we have a relatively low number of genotypes in our populations the calculation of FST values is critical and might lead to wrong results. Also for GEA studies the data basis is not suitable, but this was not the goal of the study.

Reviewer 3 Report

Dear authors,

This manuscript constitutes an interesting and well performed study concerning intraspecific diversity in aquatic ecosystems with Spirodela polyrhiza and Lemna minor. The overall idea is interesting, and the methodology is well described. The manuscript, in general, is clear, well-structured, and well written as also the results. Discussion and Conclusions should be enriched and improved. Discussion could be enriched with more relative references and the Conclusion could be as a "take home" message. 

Author Response

Dear authors,

This manuscript constitutes an interesting and well performed study concerning intraspecific diversity in aquatic ecosytems with Spirodela polyrhiza and Lemna minor. The overall idea is interesting, and the methodology is well described. The manuscript, in general, is clear, well-structured, and well written as also the results. Discussion and Conclusions should be enriched and improved. Discussion could be enriched with more relative references and the Conclusion could be as a "take home" message.

Reply: Discussion was extended by more detailed discussion of

  1. the consequences of the newly discovered hybrids between Lemna minor and other Lemna species (Braglia et al., 2021a, b)
  2. the results of S. Paolacci et al. (2021) for clones collected in South Ireland
  3. the possible transport of duckweed between ponds.

Moreover, we re-worded the Conclusion to give it more the character of a take home-message. Also the Introduction was revised and some more references were given, as suggested by the reviewer.

Round 2

Reviewer 1 Report

All my comments were suficiently addressed in a revised version. Thank you

Reviewer 2 Report

The author’s reply and revisions are in most cases satisfactory. I recommend this manuscript for publication.

Reviewer 3 Report

Dear authors,

The revised version of the manuscript is much improved.